# Effectiveness of Different Neurocognitive Intervention Approaches on Functionality in Healthy Older Adults: A Systematic Review

**DOI:** 10.3390/bs14020087

**Published:** 2024-01-25

**Authors:** Susana Sáez-Gutiérrez, Eduardo J. Fernandez-Rodriguez, Celia Sanchez-Gomez, Alberto Garcia-Martin, Luis Polo-Ferrero, Fausto J. Barbero-Iglesias

**Affiliations:** 1Department of Nursing and Physiotherapy, University of Salamanca, 37007 Salamanca, Spain; pfluis@usal.es (L.P.-F.); fausbar@usal.es (F.J.B.-I.); 2Institute of Biomedical Research of Salamanca (IBSAL), 37007 Salamanca, Spain; celiasng@usal.es; 3Department of Developmental and Educational Psychology, University of Salamanca, 37007 Salamanca, Spain; 4Department of Labour Law and Social Work, University of Salamanca, 37007 Salamanca, Spain; albergm@usal.es

**Keywords:** cognitive training, problem solving, functional limitations, activities of daily living, aging

## Abstract

Subtle loss of functionality in healthy older adults is considered one of the most important predictors of cognitive decline. Neurocognitive interventions are increasingly being used, from a preventive maintenance approach to functional capacity. This study evaluates the effectiveness of different neurocognitive approaches on the functionality of healthy older adults. In this systematic review (CRD42023473944), an extensive search was conducted for articles published in the last 10 years (2013–2023) in the following databases: Medline, Scopus, and Web of Science. A total of 809 trials were identified, of which 18 were considered to be eligible for inclusion in the review. The data revealed heterogeneity in sample size, measures of functional assessment, neurocognitive interventions used, number of sessions, session duration, and time. Traditional cognitive stimulation is shown to have no significant functional benefit, while other less commonly used neurocognitive interventions, such as those based on everyday cognition, are associated with more significant benefits. Moreover, it is demonstrated that although the Instrumental Activities of Daily Living scale (IADL) is the most used test in similar studies, it is not sensitive enough to detect changes in functionality in healthy elderly individuals, with other tests such as the Timed Instrumental Activities of Daily Living (TIADL) being more advantageous. Therefore, a new guideline is proposed for its use in clinical practice and research, using homogeneous study protocols and neurocognitive interventions that allow for the transfer and generalization of results in daily life.

## 1. Introduction

Population aging is one of the greatest health achievements, with life expectancy now reaching 72.8 years in developed countries—seven years less in less developed countries—and projected to rise to 77.2 years by 2050. However, this demographic shift is not only a triumph but also poses sustainability challenges for health and social systems [1]. In response, the World Health Organization (WHO) has published the Global Report on Ageing and Health, which asserts that healthy aging is essential to enjoying a long life of high quality, emphasizing the maintenance of health, safety, and participation in daily activities that are key to adding years to life [2].

Healthy aging is described as the functional capacity that enables well-being and is determined by intrinsic capacity (the combination of physical and mental capacity) and the physical, social, and political environment. Therefore, a multi-factorial analysis of aging that comprehends the synergy between different components is essential [2,3]. Several studies [4,5] aim to define the domains of the construct that offer a framework for guiding clinical practice and research regarding intrinsic functional capacity.

Older people with altered cognitive function are more likely to lose functional capacity and develop dementia [6]. Loss of autonomy in basic (ADLs) and Instrumental Activities of Daily Living (IADLs) is so significant that it is the discriminator between cognitively healthy older adults and a diagnosis of mild cognitive impairment or dementia [7]. It has been demonstrated that subtle functional changes precede the diagnosis of dementia by 10 to 12 years [8]. Thus, participation in everyday activities could serve as a predictor of cognitive function and, therefore, a predictor of the presence or absence of cognitive decline. One factor to consider in the daily and cognitive functioning of older adults is the cognitive reserve hypothesis. This explains why some older adults are able to maintain their cognitive abilities despite the presence of age-related brain changes or illnesses.

Furthermore, an increasing number of experts advocate for a preventive cognitive approach aimed at healthy adult individuals. The main neurocognitive intervention proposals used are “cognitive training”, “cognitive rehabilitation”, and “cognitive stimulation” [9]. Although these concepts are becoming increasingly differentiated in scientific literature, they are used interchangeably in clinical practice and healthcare, leading to confusion. Cognitive training implements guided practice of routine tasks (either individually or in groups) that reflect specific cognitive functions such as memory, attention, or problem-solving, with the aim of improving or maintaining the functioning of a particular domain [10]. Cognitive rehabilitation is a patient-centered approach that aims to identify and address the individual needs and goals of people with already established cognitive impairments, with the ultimate objective of improving everyday functioning. Cognitive stimulation involves exposure to and participation in activities and materials that require a certain degree of cognitive processing. It is usually delivered in a group setting and is more flexible than cognitive training, as it does not have specific therapeutic goals and takes into account other concepts such as social participation and affectivity [11]. Cognitive stimulation is the most commonly used concept in clinical practice [12].

The scientific literature has determined that training specific cognitive domains is effective in improving concrete cognitive functions such as working memory [13], inhibition, cognitive flexibility, selective attention [14], processing speed [15], and quality of life [16]. However, the heterogeneity of intervention programs and assessment measures remains a problem. As Tardif et al. state in a review, there is no consensus in this regard. The Mini-Mental State Examination (MMSE) remains the most used test for assessing cognitive function, despite evidence indicating its lack of sensitivity to individual differences such as level of education, age, or high-functioning individuals. This study suggests that the MoCA test is more suitable for evaluating cognitive function [17]. In addition, the number of trials on the subject remains limited, with poor methodology and lacking participant follow-up [18]. The most commonly used test in scientific literature for assessing instrumental activities of daily living is the Lawton and Brody Index, which was published in 1969. The test directly asks the person or their primary caregiver about their abilities. This test evaluates a person’s ability to use the telephone, shop, prepare food, take care of the house, do laundry, use transportation, take medication responsibly, and manage money on a scale of 0 (total dependence) to 8 (total independence).

Other more novel approaches to the use of cognitive training are new technologies, such as computer-based cognitive interventions (CCIs) [19], as they can be beneficial for preservation or improvement in healthy people or in the initial stages of cognitive disease, adding motivation and ease of adaptation. Another example is the meta-analysis conducted by Son and colleagues [20], which confirmed among its most relevant results that cognitive training based on virtual reality is a useful tool for improving IADL performance and is therefore ecologically valid.

However, the major limitation of neurocognitive intervention studies remains the problem of transferring and generalizing results to everyday functioning [21]. A meta-analysis presented in 2017 by Kim et al. on cognitive stimulation, which identified 7354 articles, highlights among its results that only three of them measured ADLs and IADLs, and none of them showed any significant benefit, coinciding with other studies that state that the main difficulty in determining the impact of cognitive interventions on everyday functioning is that they do not include measures of functional outcomes [22].

Due to the aforementioned limitations, various neurocognitive intervention proposals are being considered, including those that draw on everyday cognition, a term that refers to the capacity for solving complex cognitive problems in real-world settings. While it is true that recent research has focused on the role of fundamental cognitive skills in everyday cognitive performance [23], as demonstrated by Farias et al., who identified episodic memory, executive function, and spatial skills as the primary determinants of functional everyday skills [24], an expanding body of research [25,26,27] proposes direct intervention for everyday cognition. The 2021 RCT carried out by Sánchez et al. indicates that a daily cognitive training program has more significant advantages (in terms of global cognitive performance and everyday cognition) compared to traditional cognitive training.

Based on the above, it is necessary to conduct a comprehensive review assessing the effectiveness of various neurocognitive interventions for maintaining or improving functionality in older adults without cognitive impairment. To date, no review or meta-analysis has explored this topic. Therefore, the aim of this review is to provide a useful tool that guides research and clinical practice in selecting a cognitive training method for healthy older adults to improve or maintain daily functionality and, thus, enhance quality of life.

## 2. Materials and Methods

### 2.1. Protocol and Registration

The present systematic review was conducted following the guidelines set forth by the Preferred Reporting Items for Systematic Reviews and Meta-Analyses (PRISMA) statement, version 2020 [28]. For further details, please refer to Appendix A. The protocol was previously registered in the International Prospective Register of Systematic Reviews (PROSPERO) under the number CRD42023473944.

### 2.2. Eligibility Criteria

This research question was: “Which neurocognitive intervention approach is most effective in enhancing or maintaining functionality in healthy older adults?” The search and eligibility criteria were established through the PICO strategy [29]:

Population: The target population comprised individuals aged 60 years or older, without any diagnoses of neurodegenerative or psychiatric diseases, who participated in various neurocognitive intervention proposals.

Intervention: Any neurocognitive intervention program applied to healthy older adults as a preventive approach to functional capacity loss was eligible.

Comparison: The comparison group pertains to control groups that do not receive any intervention (passive) or receive a different treatment (active), such as usual treatments, educational talks, other interventions that differ from cognitive, etc.

Outcome: The outcome analyzed should include as one of this study variables the impact of the neurocognitive intervention on the improvement or maintenance of functioning. As there is no scientific consensus on the use of a specific questionnaire to assess functionality, it was decided to include all those standardized instruments that are frequently used in similar studies.

Scientific articles published in the last decade, from January 2013 to September 2023, in either English or Spanish were selected for analysis. Experimental clinical trials were accepted, which investigated how different proposals for neurocognitive intervention act to improve and/or maintain functionality in older adults without a diagnosis of neurodegenerative and/or psychiatric disease as one of this study variables. Exclusion criteria include pilot studies or studies without results, clinical trials without human subjects, studies with participants who were not older adults and/or did not have a clinical diagnosis of neurodegenerative and/or psychiatric disease, and studies without an objective measure of function. Studies will not be accepted if the experimental group does not receive neurocognitive intervention, either alone or combined with other pharmacological interventions (such as the use of supplements or additional medication) or non-pharmacological interventions (such as physical intervention).

### 2.3. Search Strategy

The search in the databases were conducted between July and September 2023. Medline (through Pubmed), Scopus, and Web of Science. Additionally, completed clinical trials of interest with published results were reviewed on ClinicalTrials.gov. Medical Subject Headings (MeSHs) terms used were: cognitive training, problem-solving, functional status, activities of daily living, elderly, and frail. To enhance the search sensitivity and address the ambiguity surrounding the topic in scientific literature, we incorporated additional keywords, namely functionality, cognitive stimulation, practical problem-solving, everyday problem-solving, everyday cognition, and older adults. These keywords were combined using the Boolean operators OR/AND to form the final search strategy. (“Cognitive training” OR “problem-solving” OR “cognitive stimulation” OR “practical problem solving” OR “everyday problem solving” OR “everyday cognition”) AND (“functional status” OR “activities of daily living” OR “functionality”) AND (“aged” OR “frail elderly” OR “older adults”). The search strategies in all online databases can be seen in Table 1.

### 2.4. Selection and Data Collection Processes

Authors S.S.G. and C.S.G. independently applied a process of eligibility for studies based on titles and abstracts; disagreements were discussed; and a third reviewer, E.J.F.R., was consulted when there was a lack of consensus. S.S.G. retrieved the full texts of the selected articles. Data extraction was performed by S.S.G. following the following sequence: author and year, study title, study design, sample size, age (mean and standard deviation), test/s used for functional assessment and assessment phases, description of the intervention (experimental group), passive or active group (control group/s), sessions (number, time, and duration), outcomes of interest, and quality of the studies. The characteristics of the selected studies are shown in Table 2.

### 2.5. Risk of Bias

The Physiotherapy Evidence Database (PEDro) scale [30] was applied to assess the risk of bias in the included studies. Two authors, S.S.G. and C.S.G., independently reviewed and scored the included articles according to this scale. The same authors then shared the scores and discussed them item by item. When no consensus was reached, a third author, E.J.F.R., was invited to rank them to make a final decision.

## 3. Results

### 3.1. Literature Search and Study Selection

The search yielded 809 articles for potential inclusion in the review. After removing pre-screened studies due to duplication (n = 246) and those flagged as illegible by automation tools (n = 1), the identification search resulted in 562 articles. Initial screening was carried out based on title and abstract, leaving 210 articles. Of these 210 articles retrieved for evaluation, 49 were discarded, resulting in 161 publications assessed for eligibility. Of these, 56 were not healthy older adults, 25 were not part of a clinical trial, 5 did not report outcomes, 15 did not measure function, 10 did not include neurocognitive interventions, 20 combined other physical treatments, 6 included pharmacological treatments, 5 combined other treatments, and 1 was retracted. Finally, 18 articles were selected for review. The previous selection process is outlined in the PRISMA flow diagram (Figure 1).

The quality of the selected studies, as assessed by the PEDro scale, indicates high quality, with an average score of 9.05 out of 11 across all articles.

### 3.2. Characteristics of Included Studies

In total, the 18 selected studies were published between 2014 and 2023. The total sample included 7131 individuals over the age of 60 without a clinical diagnosis of cognitive impairment. In some studies [31,32,33], older adults had subjective complaints of memory loss. The mean age of the included studies was 72 (±5) years. The studies presented widely varying figures regarding sample size, with an average size of 440 (±978). Among the studies, the smallest sample consisted of 17 participants, while the largest sample included 2912 individuals [31].

Regarding functional assessment, twelve tests were used to evaluate functionality within the selected studies. Timed Instrumental Activities of Daily Living (TIADL) [13,34,35,36], The Everyday Problem Test (EPT) [13,35,37], The Activities of Daily Living Questionnaire (ADL) [31], the Chinese version of the Observed Task of Daily Living (OTDL-C) [38], and the Clinical Dementia Rating-Sum of Boxes (CDR-SOB) [32] are all commonly used in assessing cognitive function in older adults. The IADL scale [31,39,40,41,42,43], ECB test [26,27], Barthel Index [42], OTDL tasks [37], Chula ADL index [44], EPCCE [25], and DAFS [25] were utilized. The most employed were the IADL scale and its timed version, the TIADL.

All studies included in the analysis conducted a pre- and post-intervention assessment, except for one study [37] that did not include a post-test assessment but instead conducted assessments at later time points (1–3, 5, and 10 years). Out of the included studies, five conducted follow-ups at 3 months, six at 6 months, one at 9 months, and four at 12 months. One study [35] conducted a pretest assessment, another assessment at 16 weeks, and a final assessment at the end of this study. One study [35] conducted a pretest assessment, another assessment at 16 weeks, and a final assessment at the end of this study.

Regarding the intervention of the experimental and control groups, there is significant variation between studies. The main cognitive interventions for the experimental group include cognitive stimulation programs [27,33,39,40,41,44], as well as specific programs focusing on cognitive domains such as attention [32,34], memory [33,36,37,38], and working memory [13,35], processing speed [13,34], reasoning [25,37,38], or problem-solving [39], multi-component cognitive intervention [31], and everyday cognition [26,27]. The control group had six articles with passive control, twelve with active control, and two without intervention in the control group. For the active control group, interventions consisted of alternative or habitual activities, puzzles, educational tasks, standard cognitive stimulation, memory exercises, music therapy, and mind-motor training.

On average, participants attended 28,25 (±35.24) sessions, with 5 sessions being the minimum and 144 being the maximum. Each session lasted between 30 and 120 min, with a duration ranging from 3 weeks to 9 months.

To conclude this section on functional outcomes, twelve studies show beneficial results for various neurocognitive interventions on functionality [13,25,26,27,31,34,35,36,37,38,39,41], while six studies find no significant differences [32,33,40,42,43,44].

## 4. Discussion

Active aging approaches are essential in preventing functional capacity loss and, consequently, dementia in older adults. The current systematic review aims to assess the efficacy of various neurocognitive intervention proposals on functionality. A total of 18 high-quality studies with 7131 participants were selected. The results are relevant for guiding clinical practice and research on the use of neurocognitive interventions in healthy older adults, from a preventive approach to maintaining functional status. The data suggest a high degree of heterogeneity between studies in terms of sample size, functional measures used, intervention sessions (number, timing, and duration), and outcomes.

### 4.1. Main Measures of Functional Assessment

The most commonly used scale is the Instrumental Activities of Daily Living (IADL) scale. This outcome is unsurprising as it remains the most frequently used in research, despite its demonstrated low sensitivity and biases related to gender, culture, and socio-economic level [45]. Moreover, studies have confirmed that there is no correlation between the most commonly used cognitive tests in older adults and the IADL scale [46].

A relevant finding is the association between this test and the results obtained in the selected studies, as four out of the six articles in which it was used did not show significant benefits in the main variable. The second most commonly used test was the timed version of the Instrumental Activities of Daily Living (TIADL), which showed significant effects on the primary variable in four out of five studies, making it a more sensitive measure of functionality.

### 4.2. Main Neurocognitive Interventions

Traditional cognitive stimulation remains the most utilized neurocognitive intervention. However, four out of the five studies that utilized it did not find significant benefits in functionality. This suggests, in agreement with similar studies [47], that cognitive stimulation does not provide benefits to the daily functioning of healthy older adults because the results do not transfer to everyday activities.

Among the selected articles, neurocognitive interventions that feature significant functionality benefits are those that train specific cognitive domains, such as working memory, processing speed, attention, executive functions, reasoning, or problem-solving. Additionally, interventions that incorporate everyday cognition have proven effective.

Regarding the intervention of specific cognitive domains, it makes sense for it to be beneficial in everyday functioning, as they are the foundations that enable and sustain proper performance. For example, the processing speed domain, which is related to autonomy in ADLs in various studies [48,49], or working memory.

Regarding other cognitive domains, there is more disagreement in the scientific literature on their relationship with daily performance in older adults. This is consistent with the findings of the current review, as one of the two studies that used specific neurocognitive attention interventions and one of the four studies that used memory techniques did not find significant benefits for functionality.

All studies that implemented the concept of everyday cognition in their interventions reported positive impacts on functionality. This neurocognitive approach requires individuals to utilize basic cognitive skills to resolve cognitively demanding problems, enabling autonomy in the instrumental activities of daily living. The challenges and daily activities faced by older individuals will be used as intervention material. Thus, this approach replaces specific cognitive function interventions, which have been shown in this review to not result in improved everyday functioning, with a neurocognitive intervention that uses the AIVD as both the means and the goal. Therefore, it is proposed that the most suitable neurocognitive intervention for achieving transfer and generalization of results to everyday life is everyday cognition.

### 4.3. Limitations

Due to the variability in the sessions, significant relationships with the obtained benefits have not been established, nor with the presence or type of follow-up. This confirms the need for more protocolized neurocognitive intervention modes. This prevents us from making a real comparison between the content and techniques of the different neurocognitive sessions. Regarding the high difference in the sample size, it makes it difficult to establish a real representation of this study population. This should be added to the deficits in studies we found in healthy elderly populations, as many studies that do meet the target population do not measure functionality as a study variable. This was frequently observed in the systematic review, as these two reasons were common causes for article exclusion.

One limitation of the review is that the selected studies were unable to control for all factors that may be associated with functional loss in healthy older adults, such as comorbidity, depression, or psychosocial and contextual factors that directly affect the variability of the results. Although it is true that all of them are aimed at healthy older people without cognitive impairment and that they describe and analyze sociodemographic variables such as age or education level, some are analyzed statistically.

These findings have significant implications for the early detection of cognitive decline and provide valuable information on potential targets for early intervention in healthy older adults.

## 5. Conclusions

This systematic review examines the most commonly used neurocognitive interventions currently utilized to prevent functional decline in healthy older adults. It is demonstrated that traditional cognitive stimulation has no positive effects on functionality and that new approaches should primarily aim for the transfer and generalization of results into the daily lives of healthy older individuals as a main quality of life indicator. This finding enables the direction of clinical practice and research towards new avenues, such as intervention based on everyday cognition. It lays the groundwork for future studies that should unify the use of updated assessments and interventions with current scientific literature.

## Figures and Tables

**Figure 1 behavsci-14-00087-f001:**
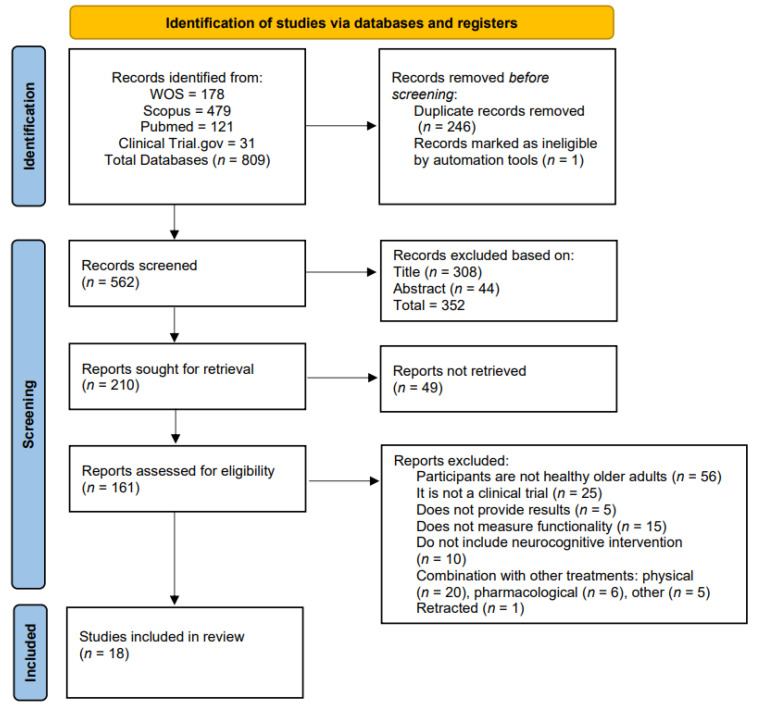
PRISMA 2020 flow diagram of systematic reviews.

**Table 1 behavsci-14-00087-t001:** Search Strategy in the Databases.

Search Strategy in the Databases
WEB OF SCIENCE TS = (“cognitive training” OR “problem solving” OR “cognitive stimulation” OR “practical problem solving” OR “everyday problem solving” OR “everyday cognition”) AND TS = (“functional status” OR “activities of daily living” OR “functionality”). AND TS = (“aged” OR “frail elderly” OR “older adults”), English or Spanish (languages), and Article or Early Access (Document Types)
SCOPUS TITLE-ABS-KEY ((“cognitive training” OR “problem solving” OR “cognitive stimulation” OR “practical problem solving” OR “everyday problem solving” OR “everyday cognition”) AND (“functional status” OR “activities of daily living” OR “functionality”) AND (“aged” OR “frail elderly” OR “older adults”)) AND PUBYEAR > 2013 AND PUBYEAR < 2023 AND (LIMIT-TO (DOCTYPE, “ar”)) AND (LIMIT-TO (LANGUAGE, “English”) OR LIMIT-TO (LANGUAGE, “Spanish”))
PUBMED (“cognitive training” OR “problem solving” OR “cognitive stimulation” OR “practical problem solving” OR “everyday problem solving” OR “everyday cognition”) AND (“functional status” OR “activities of daily living” OR “functionality”) AND (“aged” OR “frail elderly” OR “older adults”) Filters: Clinical Trial, Randomized Controlled Trial, in the last 10 years
CLINICAL TRIALS ((“aged” OR “frail elderly” OR “older adults”) AND (“functional status” OR “activities of daily living” OR “functionality”) AND (“cognitive training” OR “problem solving” OR “cognitive stimulation” OR “practical problem solving” OR “everyday problem solving” OR “everyday cognition”)) Filters: No longer looking for participants(Completed), Sex (All), Age (older adults 65+), Study Type: Interventional, Observational, Date Range: This study ran from 2013 to 2023.

**Table 2 behavsci-14-00087-t002:** Characteristics of the selected studies.

Authors and Year	Study Title	Study Design	Sample Size, Age (Mean ± SD)	Functional Outcome Measure	Intervention (Experimental Group)	Intervention (Control Group)	Sessions (Number, Time, Duration)	Results	Quality Assessment Scale (PEDro)
Belchior P., Yam A., Thomas K., Bavelier D., Ball K., Mann W., Marsiske M. (2019)	Computer and Videogame Interventions for Older Adults’ Cognitive and Everyday Functioning	Randomized Controlled Trial	Sample Size (N = 54) Mean age = 73.2 SD = 5.5	Timed Instrumental Activities of Daily Living (TIADL) Before (pretest), after (post-test), and 3 months after training	G1: Videogame (i.e., Crazy Taxi) G2: A computerized training program focused on visual attention and processing speed (i.e., PositScience InSight)	G3: Control passive	60 sessions; 1 h per session; 3 months	Both group experiments showed benefits on a measure of Timed IADL	11/11
Borella E., Cantarella A., Carretti B., De Lucia A., De Beni R. (2019)	Improving Everyday Functioning in the Old-Old with Working Memory Training	Randomized Controlled Trial	Sample Size (N = 36) Mean Age = 79 SD = 3.11	The Everyday Problem Test (EPT) Timed Instrumental Activities of Daily Living (TIADL) Before (pretest), after (post-test), and 9 months after training	G1: Working memory (WM) training	G2: Control active (alternative activities)	6 sessions; 30–40 min per session; 9-month.	The experimental group showed specific gains in the TIADL in the short term, and the follow-up showed transfer effects to everyday problem-solving (in the EPT). No such improvements were seen in the active control group	7/11
Cantarella A., Borella E., Carretti B., Kliegel M., de Beni R. (2017)	Benefits of tasks related to everyday life competences after working memory training in older adults	Randomized Controlled Trial	Sample Size (N = 36) Mean Age = 69.50 SD = 3.25	The Everyday Problem Test (EPT) and Timed Instrumental Activities of Daily Living (TIADL) Before and after the intervention	G1: Working memory (WM) training	G2: Control active (alternative activities)	5 sessions; 90 min per session; 9 months	The group experiment showed benefits and transfer effects to one of the everyday ability measures (in EPT)	7/11
Chang L., Tang Y., Chiu M., Wu C., Mao H. (2023)	A Multicomponent Cognitive Intervention May Improve Self-Reported Daily Function of Adults with Subjective Cognitive Decline	Single-arm two-period crossover trial	Sample Size (N = 17) Mean Age = 68.82 SD = 5.84	The Activities of Daily Living Questionnaire (ADL) Before (pretest), 16 weeks after baseline, preintervention, postintervention, and 16 weeks postintervention	G1: Multicomponent cognitive intervention		16 sessions; 1.5 h per session; 16 weeks	The experimental group showed significant changes from baseline to pretest (control) and pretest to posttest (intervention) on the ADLQ. Effects remained at the 16-week follow-up	5/11
Chen B., Wei Y., Deng W., Sun S. (2018)	The Effects of Cognitive Training on Cognitive Abilities and Everyday Function: A 10-Week Randomized Controlled Trial	Randomized Controlled Trial	Sample Size (N = 86) Mean Age = 68.55 SD = 5.74	Chinese version of the Observed Task of Daily Living (OTDL-C) Before and after the intervention	G1: Low ecological (LE) memory training G2: High ecological (HE) memory training G3: (LE) Reasoning Training G4: (HE) reasoning training	G5: Control passive	10 sessions; 60 min per session; 10 weeks.	The experimental groups significantly improved everyday problem-solving performance in all the intervention groups. The high ecological cognitive trainings failed to show a superior impact on everyday problem-solving compared with the low ecological cognitive trainings	9/11
Cheng C., Lam L., Cheng S. (2018)	The effects of integrated attention training for older Chinese adults with subjective cognitive complaints: A randomized controlled study	Randomized controlled trial	Sample Size (N = 93) Mean Age = 73.9 SD = (7.4)	Clinical Dementia Rating–Sum of Boxes (CDR-SOB) Before (pretest), at 3 months (post-intervention), and at 6 months	G1: The Integrated Attention Training Program (IATP)	G2: Control active (Health-related education program)	144 sessions online; 45 min per session; 3 months	The experimental group had no effect on functioning	9/11
Corbett A., Owen A., Hampshire A., Grahn J., Stenton R., Dajani S., Burns A., Howard R., Williams N., Williams Ballad C. (2015)	The Effect of an Online Cognitive Training Package on Healthy Older Adults: An Online Randomized Controlled Trial	Randomized Controlled Trial	Sample Size (N = 2912) Mean Age = 58.5 SD = 6.5	The Instrumental Activities of Daily Living scale (IADL) Before (pretest), After (post-intervention) (with additional follow-up at 6 weeks and 3 months)	G1: Problem-solving cognitive training (ReaCT) G2: General cognitive training (GCT)	G3: Control passive	10 min daily; 6 months	Both experimental groups conferred significantly greater benefit on the primary outcome measure of IADL than the control group at 6 months. Data from interim time points also shows a significant benefit to IADL at 3 months	10/11
Frankenmolen N., Overdorp E., Fasotti L., Claassen J., Kessels R., Oosterman J. (2018)	Memory Strategy Training in Older Adults with Subjective Memory Complaints: A Randomized Controlled Trial	Randomized Controlled Trial	Sample Size (N = 60) Mean Age = 66.2 SD = 7.3	The Instrumental Activities of Daily Living Scale (IADL) Before (pretest), after (post-test), and 6 months after training	G1: The memory strategy training	G2: Control active (memory training)	7 sessions; 90 min per session; 7 weeks	None of the groups (experimental and control) have found significant results or interaction effects for (IADL)	10/11
Gamito P., Oliveira J., Alves C., Santos N., Coelho C., Brito R. (2020)	Virtual Reality-Based Cognitive Stimulation to Improve Cognitive Functioning in the Community Elderly: A Controlled Study	Randomized Controlled Trial	Simple Size (N = 43) Mean Age = 75 SD = 5.43	The Instrumental Activities of Daily Living scale (IADL) Before and after the intervention	G1: Ecologically-oriented virtual reality cognitive stimulation (VR-CS)	G2: Control active (Standard cognitive stimulation (PP-CS))	G1: 12 sessions; 30 min per session; 6 weeks G2: 6 sessions; 60 min per session; 6 weeks	None of the groups (experimental and control) have found results for functionality	10/11
Gómez C., Rodríguez E. (2021)	The effectiveness of a training program in everyday cognition in healthy older adults: a randomized controlled trial	Randomized controlled trial	Simple Size (N = 237) Mean Age = 73.45 SD = 6.45	Everyday Cognition Battery Test (ECB) 8 assessments: 2 (initial and final) for each of the 4 stages of intervention	G1: Training Program in Everyday Cognition	G2: Control Active (Conventional Cognitive Training Program)	20 sessions; 50 min per session; 10 weeks	Statistically significant differences were evident between the control group and the experimental group	10/11
Gomez-Soria I., Ferreira C., Olivan-Blazquez B., Aguilar-Latorre A., Calatayud E (2023)	Effects of a cognitive stimulation program on cognition and mood in older adults, stratified by cognitive levels: A randomized controlled trial	Randomized controlled trial	Simple Size (N = 101) Mean Age and SD = Experimental group: 72.34 (0.80) Control group: 71.69 (0.77)	Barthel Index The Instrumental Activities of Daily Living scale (IADL) Before (pretest), after (post-test), and at 6- and 12-months during intervention.	G1: CS program adapted to the cognitive level		40 activities in 10 sessions; 45 min per session; 10 weeks.	In the experimental group, no significant differences were found for functionality	11/11
Gray N., Yoon J., Charness N., Boot W., Roque N., Andringa R., Harrell E., Lewis K., Vitale T. (2022)	Relative Effectiveness of General Versus Specific Cognitive Training for Aging Adults	Randomized controlled clinical trial	Sample Size (N = 230) Mean Age = 71.35 SD = 5.33	The Instrumental Activities of Daily Living scale (IADL) Before (pretest), After (post-test) 1 year after the intervention	G1: Brain training G2: Video game training G3: IADL training	G4: Control active (group with puzzle training)	40 sessions; 30 min per session; 4 weeks	No differential benefits were found in either group (experimental or control)	9/11
Rebok G., Ball K., Guey L., Jones R., Kim H., King J., Marsiske M., Morris J., Tennstedt S., Unverzagt F., Willis S. (2014)	Ten-year effects of the advanced cognitive training for independent and vital elderly cognitive training trial on cognition and everyday functioning in older adults	Randomized Controlled Trial	Sample Size (N = 2.832) Mean Age = 73.6 SD = 6.0	Everyday Problems Test (EPT) Observed Tasks of Daily Living (OTDL) Timed IADL (TIADL) Before and 1, 2, 3, 5, and 10 years after the intervention	G1: Memory training G2: Reasoning training G3: Speed-of-processing training	G4: Control passive	10 sessions; 60 to 75 min; 10 to 14 weeks	The experimental groups reported that IADL function improved over 2 years At Year 10, experimental groups reported less difficulty performing IADLs than the control group The current study showed weak to absent effects of cognitive training on performance-based measures of daily function	10/11
Rodríguez E., Gómez C., Pérez M., Iglesias F., Arenillas J. (2018)	Randomized study of an everyday cognition training program versus traditional cognitive stimulation in elderly adults	Randomized Controlled Trial	Sample Size (N = 147) Mean Age = 75.22	ECB (everyday cognition battery) Before (pretest) and after (post-intervention)	G1: Conventional Cognitive Training Program combined with Training Program in Everyday Cognition	G2: Control Active (Conventional Cognitive Training Program)	20 sessions; one hour; 10 weeks	The experimental group presents greater benefits in terms of everyday and functional cognition	10/11
Rose N., Rendell P., Hering A., Kliegel M., Bidelman G., Craik F. (2015)	Cognitive and neural plasticity in older adults’ prospective memory following training with the Virtual Week computer game	Randomized Controlled Trial	Simple Size (N = 59) Mean Age = 67 SD = 4.77	Timed Instrumental Activities of Daily Living (TIADL) Before (pretest) and after (post-intervention)	G1: Prospective memory training program using the Virtual Week computer game	G2: Control active (Music Training) G3: Control Passive	12 sessions, one hour per session; 1 month	The experimental group showed significant transfer for functional independence	5/11
Sharma S., Balaji G., Sahana A., Karthikbabu S. (2021)	Effects of Cognitive Versus Mind-Motor Training on Cognition and Functional Skills in Community-Dwelling Older Adults	Randomized Controlled Trial	Simple Size (N = 27) Mean Age = 69 SD = 4.7	The Instrumental Activities of Daily Living scale (IADL) Before (pretest) and 8 weeks after the training	G1: Cognitive Training (CT)	G2: Control active (Mind Motor Training (MMT))	24 sessions; one hour per session; 8 weeks	Both groups (experimental and control) revealed beneficial changes in IADL. The results are not significant between the groups	10/11
Srisuwan P., Nakawiro D., Chansirikarnjana S., Kuha O., Chaikongthong P., and Suwannagoot T. (2020)	Effects of Group-Based 8-Week Multicomponent Cognitive Training on Cognition, Mood, and Activities of Daily Living among Healthy Older Adults: A One-Year Follow-Up of a Randomized Controlled Trial	Randomized Controlled Trial	Simple Size (N = 77) Mean Age = 65.7 SD = 4.3	The Chula ADL index Before (pretest), 6 months and 1 year after training	G1: Training program of executive functions, attention, memory, and visuospatial functions (TEAM-V Program)	G2: Control active (Usual treatment)	5 sessions; 120 min per session; 10 week	The experimental group has not recorded significant results in functionality	9/11
Williams K., Herman R., Bontempo D. (2014)	Reasoning Exercises in Assisted Living: a cluster randomized trial to improve reasoning and everyday problem solving	Randomized Controlled Trial	Sample Size (N = 89) Mean Age = 86 SD = 5.9	Every Day Problems Test for Cognitively Challenged Elders (EPCCE) Direct Assessment of Functional Status (DAFS) Before (pretest) After (post-test) and 3-months and 6-months after intervention	G1: Reasoning Exercises in Assisted Living (REAL)	G2: Control active (vitamin/nutrition-education program (VITAMIN)) G3: Control passive	6 sessions; 3 weeks	The experimental group showed significant increases in the two functionality tests	11/11

## Data Availability

Please contact the corresponding author to request data.

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
