# Peer review of "Effectiveness of Different Neurocognitive Intervention Approaches on Functionality in Healthy Older Adults: A Systematic Review"

_behavsci, 2024, doi:10.3390/bs14020087_

Round 1

Reviewer 1 Report

Comments and Suggestions for Authors

This article presents a contribution to the literature by shedding light on the nuanced effectiveness of neurocognitive interventions on functional capacity in healthy older adults. The study's focus on evaluating the effectiveness of various neurocognitive approaches in maintaining functional capacity. Here are some concerns:

1.In the realm of cognitive research, antecedent studies have divulged compelling evidence suggesting that neurocognitive intervention holds the potential to augment cognitive reserve. For the demographic encompassing both middle-aged and elderly individuals, neurocognitive intervention can be delineated into various typologies, notably bilingualism, lifelong learning, physical activity, engagement in leisure time pursuits, and cultivation of expansive social networks. Each of these distinct categories imparts a unique influence on cognitive function. The efficacy and potency of interventions vary across cognitive spheres. The inadequacy becomes conspicuous when one attempts to extrapolate from generic interventions without delving into the specific subcategories encapsulated within neurocognitive interventions. 

Ref:

van Os Y, de Vugt ME, van Boxtel M. Cognitive Interventions in Older Persons: Do They Change the Functioning of the Brain? Biomed Res Int. 2015;2015:438908.

Chan D, Shafto M, Kievit R, Matthews F, Spink M, Valenzuela M; Cam-CAN; Henson RN. Lifestyle activities in mid-life contribute to cognitive reserve in late-life, independent of education, occupation, and late-life activities. Neurobiol Aging. 2018 Oct;70:180-183.

Jin Y, Lin L, Xiong M, Sun S, Wu SC. Moderating effects of cognitive reserve on the relationship between brain structure and cognitive abilities in middle-aged and older adults. Neurobiol Aging. 2023 Aug;128:49-64.

2.  A visual representation of the screening procedure would enhance transparency and facilitate a clearer comprehension of the rigorous methodology applied in this extensive literature review. The authors are encouraged to furnish a specific flowchart elucidating the systematic selection process employed to distill 18 articles from an initial pool of over 800. 

3. The Instrumental Activities of Daily Living is most widely used measure, and the authors are expected to provide a comprehensive exposition elucidating the specific components of this pivotal assessment tool.

4. What is the term "Traditional cognitive stimulation" refers to in this article?

5. The overall structure and logical relationships within the discussion section are not particularly clear. It is recommended that the author introduce subheadings in the discussion section to enhance clarity and organization.

6. The discussion section tends to be superficial, lacking in-depth exploration of specific neurobiological mechanisms.

Reviewer 2 Report

Comments and Suggestions for Authors

See review file

Round 2

Reviewer 1 Report

Comments and Suggestions for Authors

The title of the paper encompasses different neurocognitive interventions, with the author attempting to provide a comprehensive review of neurocognitive intervention. However, the overall coverage of neurocognitive interventions in the article is relatively limited, and this limitation is closely tied to the use of keywords. Many neurocognitive interventions, such as Neurofeedback training, Cognitive Remediation Therapy, Mindfulness-Based Interventions, Transcranial Magnetic Stimulation, Computerized Cognitive Rehabilitation, Physical Exercise Interventions, Social Cognitive Training, and others, are not included in the search keywords.

The design principles of these neurocognitive interventions are grounded in relevant discoveries in neuroscience and corresponding theoretical models. Therefore, the author may choose to either employ a more inclusive set of keywords during the search process, aiming to present readers with a comprehensive perspective; or initiate the discussion from pertinent findings and theoretical models within the field of neuroscience, outlining the fundamental design principles of neurocognitive interventions, delving into a more comprehensive examination of neurocognitive interventions.

Round 3

Reviewer 1 Report

Comments and Suggestions for Authors

The author has provided thorough responses to the raised inquiries. It is recommended that the work be considered for publication.